# CausalBench: A Comprehensive Benchmark for Evaluating Causal Reasoning Capabilities of Large Language Models

## Abstract

Causal reasoning, a core aspect of human cognition, is essential for advancing large language models (LLMs) towards artificial general intelligence (AGI) and reducing their propensity for generating hallucinations. However, existing datasets for evaluating causal reasoning in LLMs are limited by narrow domain coverage and a focus on cause-to-effect reasoning through textual problems, which does not comprehensively assess whether LLMs truly grasp causal relationships or merely guess correct answers. To address these shortcomings, we introduce a novel benchmark that spans textual, mathematical, and coding problem domains. Each problem is crafted to probe causal understanding from four perspectives: cause-to-effect, effect-to-cause, cause-to-effect with intervention, and effect-to-cause with intervention. This multi-dimensional evaluation method ensures that LLMs must exhibit a genuine understanding of causal structures by correctly answering questions across all four dimensions, mitigating the possibility of correct responses by chance. Furthermore, our benchmark explores the relationship between an LLM's causal reasoning performance and its tendency to produce hallucinations. We present evaluations of state-of-the-art LLMs using our benchmark, providing valuable insights into their current causal reasoning capabilities across diverse domains. The dataset is publicly available for download at https://huggingface.co/datasets/CCLV/CausalBench.

## 1  Introduction

Causal reasoning, the ability to understand and infer causal relationships between variables, is a fundamental aspect of human cognition and plays a crucial role in decision-making, problem-solving, and learning [1]. For large language models (LLMs), causal reasoning refers to the ability to accurately identify, represent, and reason about causal relationships described in text, mathematical equations, or code snippets [1]. Developing strong causal reasoning abilities in LLMs is essential for progress toward artificial general intelligence (AGI), as it enables models to understand not just correlations but the underlying mechanisms driving outcomes [3]. This understanding is crucial for making accurate predictions, generating insightful explanations, and adapting to new situations, as core components of AGI.

However, existing causal reasoning benchmarks have several limitations that hinder their ability to comprehensively evaluate the causal reasoning capabilities of LLMs. First, current benchmarks often focus on a single perspective of causal reasoning, such as cause-to-effect, lacking a multifaceted assessment that considers effect-to-cause reasoning and the impact of interventions. This narrow focus allows models to correctly answer causal questions by chance without truly understanding the underlying causal relationships [5]. Second, current benchmarks are primarily text-based, lacking diversity in problem types, such as mathematical and coding problems that can encapsulate causal dependencies. Incorporating these diverse problem formats would enable a more robust evaluation

Submitted to the 38th Conference on Neural Information Processing Systems (NeurIPS 2024) Track on Datasets and Benchmarks. Do not distribute.

of LLMs' capacity to reason about causality across various modalities. Third, the limited scale of existing benchmarks may not provide a sufficiently comprehensive assessment of LLMs' causal reasoning abilities due to the limited scale of the benchmark dataset.

To address these limitations, we propose CausalBench, a comprehensive benchmark for evaluating the causal reasoning capabilities of LLMs. CausalBench comprises four perspectives of causal reasoning for each scenario: cause-to-effect, effect-to-cause, cause-to-effect with intervention, and effect-to-cause with intervention. This multi-perspective approach mitigates the potential for correct answers by chance and provides a more accurate evaluation of LLMs' understanding of causal relationships. Moreover, CausalBench includes a diverse set of problem types spanning textual, mathematical, and coding domains, enabling a comprehensive assessment of causal reasoning abilities across different modalities. The benchmark consists of more than 60,000 problems and employs six evaluation metrics to measure LLMs' causal reasoning performance.

The major contributions of CausalBench are three-fold: (1) evaluating four causal reasoning perspectives per scenario to robustly assess causal understanding, (2) incorporating a diverse problem set spanning math, code, and natural language for cross-modal evaluation, and (3) implementing strict quality control measures, including a causal inference engine check and human expert review, to ensure the benchmark's validity and reliability. By addressing the limitations of existing benchmarks, CausalBench aims to provide a more comprehensive and accurate evaluation of the causal reasoning capabilities of LLMs, facilitating progress towards AGI.

## 2 Related Works

Existing datasets and benchmarks for evaluating causal reasoning primarily focus on commonsense causality [9, 31, 32], which assesses the alignment between commonsense knowledge about causal relationships in humans and language models. These datasets, such as WikiWhy [9], CausalWorld [31], and UCLM [32], provide valuable insights into how well language models capture and reason about everyday causal relationships. However, they do not explicitly evaluate the models' ability to perform formal causal reasoning based on well-defined rules and principles from the field of causal inference. Some recent works have started to explore more formal aspects of causal reasoning in language models. For example, CRASS [28] focuses specifically on counterfactual reasoning, which involves reasoning about alternative outcomes based on hypothetical changes to past events. While counterfactual reasoning is an important aspect of causal inference, CRASS does not cover the full spectrum of causal inference tasks, such as interventional and observational reasoning. Another concurrent work by Kiciman et al. [16] evaluates language models on various causality-related tasks, including causal sufficiency analysis, causal discovery, and counterfactual reasoning. However, their evaluation primarily relies on the conceptual knowledge accrued from the training data rather than formal causal inference, except for their causal sufficiency analysis. This means that the models' performance may be influenced by spurious correlations or memorization from the training data rather than a genuine understanding of causal principles.

In contrast, our proposed dataset, CausalBench, is grounded in the principles of causal inference [11, 25, 26]. CausalBench provides a comprehensive and principled framework for assessing the causal reasoning capabilities of language models, ensuring that the models are evaluated on their ability to perform formal causal inference rather than relying on spurious correlations or memorization from training data. By encompassing a diverse set of causal scenarios (text, code, and math), four causal perspectives (cause to effect, effect to cause, cause to effect with intervention, and effect to cause with intervention), and explanations associated with ground truth for each test case, CausalBench offers a rigorous and systematic approach to benchmarking causal reasoning in LLMs. It is designed to test the models' ability to reason about causal relationships in a variety of domains, including natural language, programming code, and mathematical equations. In summary, while existing datasets and benchmarks have made contributions to the study of causal reasoning in language models, CausalBench offers a more comprehensive, principled, and rigorous approach to evaluating formal causal inference capabilities across multiple domains. By grounding the evaluation in the principles of causal inference and providing a diverse set of test cases with associated explanations, CausalBench aims to set a new standard for benchmarking causal reasoning in LLMs.

# 3 Dataset Construction Process and Method

The construction of CausalBench involves three key steps: manual generation of initial test cases, scaling up using LLM such as GPT-4 Turbo, and quality control through causal inference engines together with human verification. Initially, we manually create a set of test cases covering four aspects of causal inference: (a) cause to effect, (b) effect to cause, (c) cause to effect with intervention, and (d) effect to cause with intervention to ensure a comprehensive evaluation of causal reasoning capabilities from different perspective. To expand the dataset, we then use GPT-4 Turbo with few-shot prompting, leveraging the model's ability to generate additional test cases that adhere to the desired format and cover the four causal inference aspects. The few-shot prompts are designed to guide GPT-4 Turbo in producing a diverse and extensive set of problems that maintain consistency with the manually generated cases. Afterward, we implement a quality control process involving validation through causal inference engines and review by human experts. The causal inference engines verify the logical consistency and correctness of the generated test cases, while human experts review and refine the dataset to maintain high standards of quality and relevance.

## 3.1 Workflow Overview

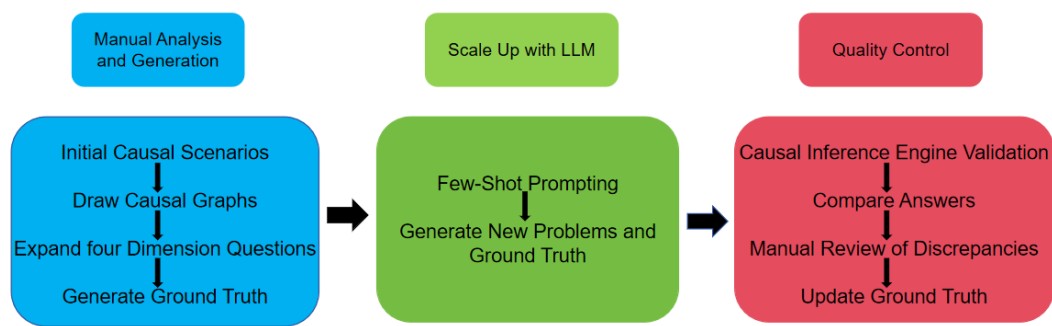

Figure 1: Workflow overview of the CausalBench dataset construction process.

## 3.2 Manual Analysis and Generation

For the text problems of our Benchmark, we randomly selected 100 questions from the CLADDER dataset [10] and manually analyzed them to determine their category within (1) inference from cause to effect, (2) effect to cause, (3) cause to effect with intervention, or (4) effect to cause with intervention. These perspectives represent different dimensions of causal reasoning: (1) Cause to the effect: Given the cause, what is the likelihood of the effect? (2) Effect to cause: Given the effect, what is the likelihood of the cause? (3) Cause to effect with intervention: If an intervention is added to the causal relationship, given the cause, what is the likelihood of the effect? and (4) Effect to cause with intervention: If an intervention is added to the causal relationship, given the effect, what is the likelihood of the cause?

After categorizing the selected cases from the CLADDER dataset, we expanded them by creating additional questions for the other three perspectives. For example, if a case was classified as "cause to effect", we generated corresponding questions for "effect to cause", "cause to effect with intervention", and "effect to cause with intervention" manually.

To correctly expand other perspective questions and their ground truths, we visualized the relationships between variables using causal diagrams and analyzed these relationships by calculating conditional probabilities. Causal diagrams represent variables as nodes and causal relationships as directed edges. For example, consider the following hypothetical scenario:

*Imagine a self-contained, hypothetical world with only the following conditions, and without any unmentioned factors or causal relationships: Parents' intelligence has a direct positive effect on parents' social status and child's intelligence. Other unobserved factors has a positive direct effect on parents' social status and child's intelligence. If a child is intelligent, would it be more likely that this child had intelligent parents?*

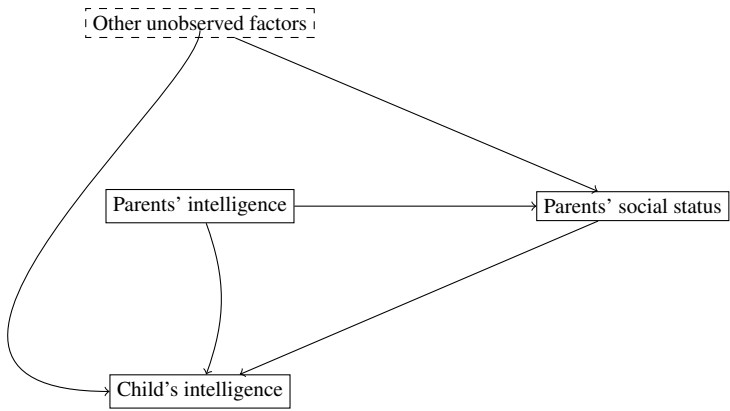

Figure 2: Causal Graph Example

In this scenario, the causal diagram would have four nodes: Parents' intelligence, Parents' social status, Child's intelligence, and Other unobserved factors. There would be directed edges from Parents' intelligence to Parents' social status and Child's intelligence, from Other unobserved factors to Parents' social status and Child's intelligence, and from Parents' social status to Child's intelligence. Conditional probabilities can be estimated based on the causal graph.

Using the causal graph and conditional probabilities, we can categorized the original questions as effect-to-cause. The probability of the child being intelligent given that the parents are intelligent is higher than the probability of the child being intelligent given that the parents are unintelligent, so the ground truth is yes. Then extend the questions to cover four perspectives by adjusting the questioning logic and incorporating interventions into the causal path diagram, and calculate ground truth for each questions.(examples are provided in the Appendix)

Finally, we obtained 100 causal scenarios, with 400 causal questions. They serve as the foundation for our few-shot prompting approach, providing examples for GPT-4 Turbo on how to identify the type of the initial question and generate additional questions for the remaining perspectives. By using these examples in a few-shot prompting setting, we guide the model to generate additional perspective questions with answers for all other causal scenarios in the CLADDER dataset.

For coding and mathematical problems, we manually created 100 code scenarios and 100 math scenarios, each containing causal relationships, and designed four perspective questions for each scenario. These questions addressed causal issues based on the relationships described in the scenarios (examples are provided in the Appendix). We then used causal graphs and conditional probabilities to manually generate the ground truths and employed few-shot prompts with GPT-4 Turbo to generate additional code, math scenarios and questions with corresponding answers.

In summary, the manual analysis and generation process involved visualizing causal relationships using causal diagrams and calculating conditional probabilities for each scenario. We modified the questioning approach and added interventions to expand each problem into four forms, covering cause-to-effect, effect-to-cause, cause-to-effect with intervention, and effect-to-cause with intervention, and generated ground truths for each question. By the end of this section, we had created 100 sets of 400 text-based questions with ground truths, 100 sets of 400 coding questions with ground truths, and 100 sets of 400 math questions with ground truths. These manually generated samples serve as the foundation for our few-shot prompting approach, which utilizes GPT-4 Turbo to generate additional test cases.

## 3.3 Scaling Up with LLMs

After manually generating and verifying an initial set of questions, we employed GPT-4 Turbo to scale up the dataset. The scale-up process was divided into three parts: text problems, coding problems, and mathematical problems.

For the text problems, we provided GPT-4 Turbo with original CLADDER dataset[10] questions with manually expanded questions along with their ground truths. By learning from these samples, GPT-4

Turbo was tasked with reading the remaining CLADDER scenarios (around 10,000 problems) and their corresponding questions, determining the question perspective, expanding the scenario into the other three perspectives, and generating the associated ground truths. This process ensures every text causal scenario has four dimension questions and corresponding ground truths.

In the case of coding problems, we supplied GPT-4 Turbo with the 100 manually created code examples containing causal relationships. Using these examples as a foundation, GPT-4 Turbo generated an additional 2,000 code snippets, each incorporating causal relationships. For each newly generated code snippet, GPT-4 Turbo created four perspectives of questions and provided the corresponding ground truths, ensuring a comprehensive evaluation of causal reasoning in the context of programming.

Similarly, for mathematical problems, GPT-4 Turbo was employed to generate 2,000 new mathematical scenarios across various domains, such as probability theory, mathematical statistics, differential equations, and complex analysis. For each mathematical scenario, GPT-4 Turbo generated four types of questions and their associated ground truths, assessing the model's ability to reason about causal relationships in mathematical contexts.

By leveraging the capabilities of GPT-4 Turbo, we were able to create a dataset across all three problem categories. The text problems were augmented by automatically generating additional question perspectives and ground truths based on the existing CLADDER scenarios. The coding and mathematical problems were scaled up by having GPT-4 Turbo create new scenarios containing causal relationships and generate the corresponding questions and ground truths. This scale-up process resulted in a more comprehensive and diverse dataset, enabling a thorough evaluation of causal reasoning abilities in large language models across various domains.

## 3.4 Quality Control

### 3.4.1 Causal Inference Engine Design

To ensure the accuracy and consistency of the generated questions and answers, we developed a causal inference engine. This engine utilizes causal diagrams and conditional probabilities associated with each question to compute the answers for all questions. The causal inference engine serves as a verification layer, comparing the answers generated by the language model. If the answer generated by the language model differs from the answer generated by the causal inference engine, the case will be manually inspected, and the ground truth will be generated by human experts. Here are the Causal Inference Engine design details:

**Input**

- A causal scenario described in natural language, code, or mathematical equations, including causal relationships among variables, known conditions, etc.
- A causal query, which is a question based on causal scenario

**Steps**

1. **Causal Graph Extraction:**
   (a) For natural language scenarios, we identify variables and causal relationships, and construct causal graphs ($G := (V, E)$) by implementing a pipeline consisting of semantic parsing and coreference resolution modules. The semantic parsing module first uses the Stanford Parser [12] to perform syntactic parsing and obtain the sentence structure. Then, it applies Compositional Semantics [13] to recursively map the syntactic parse tree to a logical form, based on the principle of compositionality. The coreference resolution module uses techniques such as the mention-pair model [14] to determine which mentions refer to the same entity, and merges the variables corresponding to coreferent mentions. From the outputs of the semantic parsing and coreference resolution modules, the pipeline automatically extracts variables from nouns and noun phrases, and identifies causal relationships indicated by verbs and conjunctions expressing causality [15]. Finally, the causal graph construction module takes the extracted variables as nodes (V) and causal relationships as directed edges (E) to automatically build the causal graph [1].

(b) For code scenarios, we identify variables and their dependencies, and construct causal graphs by implementing a pipeline that analyzes the code structure, control flow, and data flow. The pipeline first uses a code parser, such as the ast module [17] in Python, to generate an abstract syntax tree (AST). It then performs control flow analysis using techniques like control flow graphs (CFGs) [18] and program dependence graphs (PDGs) [21], and data flow analysis using def-use chains [19] and static single assignment (SSA) form [20], to identify execution paths, dependencies between statements, and variable dependencies. These analyses help automatically extract variables and their relationships from the code structure. Finally, the causal graph construction module takes the extracted variables as nodes (V) and their dependencies as edges (E) to build the causal graph based on the code semantics [1], capturing the causal relationships between variables and enabling further reasoning and analysis.

(c) For math scenarios, we identify variables and their functional relationships, and construct causal graphs by implementing a pipeline that parses and analyzes the mathematical equations. The pipeline first uses a math expression parser, such as the SymPy library [22] in Python, to convert the equations into an abstract syntax tree (AST) representation. It then traverses the AST to identify variables and their functional relationships, such as dependencies and algebraic operations, using techniques like symbolic differentiation [23] and expression simplification [24]. These analyses help automatically extract variables and their relationships from the equation structure. Finally, the causal graph construction module takes the extracted variables as nodes (V) and their functional relationships as directed edges (E) to build the causal graph based on the equation semantics, similar to the approach in [1]. The resulting causal graph captures the causal relationships between variables in the mathematical equations, enabling further reasoning and analysis.

2. **Query Classification:** Classify the causal query into one of the three levels of the Ladder of Causation (Association, Intervention, Counterfactuals). Formalize the query into the corresponding causal language, as discussed in [4].

3. **Estimand Derivation:**

   (a) For text and math scenarios, we construct a module that uses causal inference algorithms (e.g., do-calculus [25], counterfactual inference formulas [26]) to derive the estimand based on the causal graph and query type.

   (b) For code scenarios, we use program analysis techniques (e.g., symbolic execution, data dependency analysis, control flow analysis) to derive the estimand based on the code structure and query type. This involve simulating interventions on code variables and analyzing the resulting program behavior.

4. **Data Matching:** Match the terms in the estimand with the available data or constraints in the scenario to obtain a computable estimand expression. Check the completeness and consistency of the data. Raise warnings or errors if critical data is missing. For code scenarios, this involve executing the code with specific inputs and observing the outputs. This step is similar to the data matching phase in [4].

5. **Causal Effect Estimation:**

   (a) Calculate the causal effect value based on the estimand expression and the available data, yielding the answer to the query.

   (b) For scenarios with unobserved confounders, use instrumental variable estimation [27] or front-door adjustment [25].

   (c) For code scenarios, this involve comparing program behaviors under different interventions.

   This step is inspired by causal effect estimation phase in [4].

**Output**

- Answer to the causal query, including the estimated causal effect, confidence interval, and key assumptions.

In a summary, our Causal Inference Engine extends the original design presented in [4] by incorporating domain-specific graph extraction and estimand derivation techniques to handle causal inference problems in text, code, and math scenarios. The overall pipeline remains consistent with the one described in [4], but the internal methods are adapted to the specific structures and semantics of each domain.

### 3.4.2 Quality Control Process

After expansion with GPT4-Turbo, we obtained around 10000 x 4 text-based questions, 2000 x 4 math questions, and 2000 x 4 coding questions, along with their GPT-4 Turbo generated answers. To ensure the accuracy of the ground truth of each questions, we employed a strict quality control process as showing below:

We used the causal inference engine introduced above to independently solve the problems and generate its own set of answers. We compared the answers generated by GPT-4 Turbo and the causal inference engine. If two answers were the same, we updated the answer as ground truth. If any of the answers were inconsistent, we conducted a manual analysis of the question and answers to determine the correct answer and update ground truth accordingly.

This multi-step quality control process, involving the use of causal inference engine and human expert check, ensures that the final dataset contains accurate and reliable questions and answers. The manual review of inconsistent answers further enhances the quality of the dataset by addressing any discrepancies or edge cases that the models may encounter.

## 4 Benchmark Results

### 4.1 Baseline of Mainstream LLMs

We tested several state-of-the-art large language models, including GPT-4, Claude-3, LLAMA-3, and others, on our CausalBench. The evaluation metrics included: Four-Type Questions Group Correction Rate, Overall Correction Rate (Ignore Question Type), From Cause to Effect without Intervention Correction Rate, From Effect to Cause without Intervention Correction Rate, From Cause to Effect with Intervention Correction Rate, and From Effect to Cause with Intervention Correction Rate. For each causal scenario, there are four questions: cause-to-effect without intervention, effect-to-cause without intervention, cause-to-effect with intervention, and effect-to-cause with intervention. The Four-Type Questions Group Correction Rate represents the proportion of scenario cases where all four types of questions of one scenario are all answered correctly by the large language models. If any of the four questions of a scenario is answered incorrectly, the scenario is considered to be answered incorrectly by the LLM. The Overall Correction Rate (Ignore Question Type) is calculated by dividing the total number of correctly answered questions by the total number of questions, without categorizing the questions by type and scenario. The From Cause to Effect without Intervention Correction Rate is calculated by dividing the number of correctly answered "From Cause to Effect without Intervention" type questions by the total number of this type of questions. Similarly, the From Effect to Cause without Intervention Correction Rate is calculated by dividing the number of correctly answered "From Effect to Cause without Intervention" type questions by the total number of this type of questions. The remaining two metrics, From Cause to Effect with Intervention Correction Rate and From Effect to Cause with Intervention Correction Rate, follow the same calculation method as the previous two metrics, focusing on their respective question types.

Here are the tables showing LLMs' performance on text, math, and code problems.

| Model | Four-Type Questions Group Correction Rate (%) | Overall Correction Rate (Ignore Question Type) (%) | From Cause to Effect without Intervention Correction Rate (%) | From Effect to Cause without Intervention Correction Rate (%) | From Cause to Effect with Intervention Correction Rate (%) | From Effect to Cause with Intervention Correction Rate (%) |
|---|---|---|---|---|---|---|
| GPT-4 Turbo | 36.9 | 73.3 | 74.4 | 71.2 | 73.8 | 73.7 |
| Claude3-Opus | 36.8 | 72.6 | 74.1 | 70.9 | 73.2 | 72.2 |
| Mistral-7B | 25.5 | 63.6 | 58.7 | 66.5 | 64.2 | 65.0 |
| Llama3-70B | 21.8 | 61.5 | 62.6 | 59.6 | 63.8 | 60.1 |
| Llama2-7B | 20.7 | 62.1 | 62.8 | 64.0 | 56.4 | 65.4 |
| GPT-3.5 | 16.7 | 57.8 | 57.6 | 58.5 | 56.2 | 58.7 |
| Gemma-7b-it | 12.8 | 50.7 | 50.0 | 46.9 | 53.6 | 52.1 |
| Bloomz | 4.2 | 41.7 | 41.0 | 40.7 | 41.7 | 43.6 |
| AquilaChat | 1.9 | 31.1 | 28.7 | 32.4 | 33.1 | 30.4 |

Table 1: LLM Performance on Text Problems

| Model | Four-Type Questions Group Correction Rate (%) | Overall Correction Rate (Ignore Question Type) (%) | From Cause to Effect without Intervention Correction Rate (%) | From Effect to Cause without Intervention Correction Rate (%) | From Cause to Effect with Intervention Correction Rate (%) | From Effect to Cause with Intervention Correction Rate (%) |
|---|---|---|---|---|---|---|
| Mistral-7B | 62.0 | 87.2 | 78.9 | 85.6 | 85.3 | 98.9 |
| GPT-4 Turbo | 61.4 | 88.7 | 78.6 | 88.3 | 91.7 | 96.0 |
| Claude3-Opus | 54.6 | 85.9 | 74.7 | 87.1 | 86.5 | 95.4 |
| Llama3-70B | 40.8 | 80.7 | 56.8 | 86.8 | 82.0 | 97.1 |
| Gemma-7b-it | 38.3 | 79.2 | 50.4 | 82.8 | 91.1 | 92.0 |
| AquilaChat | 25.3 | 68.1 | 57.0 | 67.8 | 69.2 | 78.3 |
| Bloomz | 23.9 | 69.2 | 53.3 | 76.8 | 67.3 | 79.7 |
| GPT-3.5 | 15.9 | 63.3 | 47.1 | 71.5 | 48.6 | 86.1 |
| Llama2-7B | 2.8 | 42.3 | 45.3 | 54.2 | 17.5 | 52.4 |

Table 2: LLM Performance on Math Problems

| Model | Four-Type Questions Group Correction Rate (%) | Overall Correction Rate (Ignore Question Type) (%) | From Cause to Effect without Intervention Correction Rate (%) | From Effect to Cause without Intervention Correction Rate (%) | From Cause to Effect with Intervention Correction Rate (%) | From Effect to Cause with Intervention Correction Rate (%) |
|---|---|---|---|---|---|---|
| Llama3-70B | 43.8 | 77.0 | 82.0 | 75.7 | 73.9 | 76.0 |
| Claude3-Opus | 39.6 | 71.3 | 78.6 | 71.3 | 68.7 | 66.5 |
| GPT-4 Turbo | 37.2 | 71.0 | 80.6 | 67.5 | 73.2 | 62.5 |
| Gemma-7b-it | 32.3 | 68.4 | 74.1 | 67.7 | 66.0 | 65.4 |
| Mistral-7B | 31.4 | 66.8 | 67.5 | 68.3 | 61.3 | 70.2 |
| GPT-3.5 | 25.0 | 64.5 | 71.9 | 65.4 | 59.8 | 60.6 |
| Llama2-7B | 22.6 | 61.9 | 79.0 | 45.5 | 76.3 | 46.8 |
| Bloomz | 17.5 | 52.4 | 49.6 | 56.8 | 46.4 | 56.8 |
| AquilaChat | 14.7 | 47.3 | 36.8 | 56.4 | 38.9 | 57.2 |

Table 3: LLM Performance on Code Problems

## 4.2 Test Result Summary

The evaluation results of state-of-the-art large language models on CausalBench provide valuable insights into their causal reasoning capabilities across textual, mathematical, and coding problem domains:

Overall, the models achieved higher correction rates on mathematical problems compared to textual and coding problems. For instance, GPT-4 achieved an 88.7% overall correction rate on math

problems, while scoring 73.3% and 71.0% on text and code problems, respectively. This suggests that causal reasoning in mathematical contexts is relatively easier for LLMs compared to natural language and programming domains.

The Four-Type Questions Group Correction Rate, which measures the proportion of scenarios where all four reasoning perspectives are correctly answered, was consistently lower than the Overall Correction Rate (Ignore Question Type) across all problem types. For example, GPT-4 achieved a 61.4% Four-Type Questions Group Correction Rate on math problems, compared to an 88.7% Overall Correction Rate. This indicates that LLMs often struggle to maintain a comprehensive understanding of causal relationships when questioned from multiple perspectives.

The introduction of interventions in the causal scenarios led to mixed results in correction rates across models and problem types. In the text domain, the correction rates slightly decreased for most models when interventions were introduced. However, in the math domain, the correction rates generally improved with interventions. For instance, GPT-4's performance increased from 78.6% to 91.7% on cause-to-effect questions with intervention in math problems. In the coding domain, the impact of interventions varied across models, with some showing improvements and others exhibiting a decline in performance.

Among the tested models, GPT-4 and Claude-3 consistently outperformed other large language models (LLMs) across most problem types and reasoning dimensions, achieving the highest correction rates. Mistral demonstrated strong performance in mathematical problems but exhibited shortcomings in code-related tasks. Conversely, LLAMA-3 showed robust performance in code-related problems but faced challenges with text and mathematical tasks.

## 5 Correlation with Hallucination

To analyze the correlation between LLMs' causal reasoning ability and their hallucination rate, we referred to the LLMs' performance on hallucination datasets. The hallucination evaluation results were obtained from the Hallucination Leaderboard, developed by Vectara [30]. This leaderboard provides a comparison of LLM performance in maintaining a low hallucination rate and ensuring factual consistency when summarizing a set of facts.

| Model | Hallucination Rate | Factual Consistency Rate | Answer Rate | Average Summary Length (Words) |
|---|---|---|---|---|
| GPT 4 Turbo | 2.5 % | 97.5 % | 100.0 % | 86.2 |
| Llama3-70B | 4.5 % | 95.5 % | 99.2 % | 68.5 |
| Mistral 7B Instruct-v0.2 | 4.5 % | 95.5 % | 100.0 % | 106.1 |
| Llama2-7B | 5.6 % | 94.4 % | 99.6 % | 119.9 |
| Claude3-Opus | 7.4 % | 92.6 % | 95.5 % | 92.1 |
| Google Gemma-7b-it | 7.5 % | 92.5 % | 100.0 % | 113.0 |

Table 4: Performance of LLMs on the Hallucination Dataset

The hallucination evaluation process involves measuring the hallucination rate, factual consistency rate, answer rate, and average summary length. These metrics provide a comprehensive understanding of each model's tendency to hallucinate and its ability to maintain factual accuracy [30].

After comparing the LLMs' performance on CausalBench with their performance on the Hallucination evaluation leaderboard provided by Vectara on Huggingface [30], we found that models with stronger causal reasoning abilities tend to exhibit lower hallucination rates. For instance, GPT-4 Turbo, LLAMA-3-70B, and Mistral-7B, which demonstrated superior performance on causal reasoning tasks, also had low hallucination rates. In contrast, models like Google Gemma-7b-it and LLAMA-2-7B, which showed weaker performance on our CausalBench, had higher hallucination rates of 7.5% and 5.6%, respectively.

This trend indicates a potential link between a model's ability to understand and reason about causal relationships and its likelihood of not producing hallucinations. Further research is required to explore this correlation in more depth and to understand the underlying mechanisms driving this relationship.

## 6 Impact and Limitations

### 6.1 Impact

For the first time, we innovatively propose four types of questioning approaches for the same causal scenario: cause-to-effect, effect-to-cause, cause-to-effect with intervention, and effect-to-cause with intervention. We also calculate the proportion of cases where large language models correctly answer all four types of questions for a given causal scenario. This effectively avoids the situation where large language models coincidentally answer causal questions correctly without understanding the causal relationships embedded in the causal scenario, thereby improving the accuracy of the dataset's test results. By providing causal reasoning problems spanning multiple domains(text, code, math), it addresses the limitations of existing causal datasets and offers a more comprehensive and robust tool for assessing the causal reasoning abilities of language models. The findings in this paper suggest that models with stronger causal reasoning capabilities tend to exhibit lower hallucination rates, providing a new perspective on exploring the relationship between causal reasoning and reducing hallucinations. CausalBench has the potential to become a benchmark for driving progress in causal reasoning in artificial intelligence.

### 6.2 Limitations

CausalBench has several limitations that need to be addressed in future work. These include the need for further expanding the domain coverage, increasing the scale of the dataset, incorporating causal discovery tasks and exploring the intrinsic mechanisms between causal reasoning and hallucinations through more empirical studies.

## 7 Conclusion

In this paper, we present CausalBench, a comprehensive benchmark dataset for evaluating the causal reasoning capabilities of large language models. CausalBench innovatively proposes four types of questioning approaches for each causal scenario: cause-to-effect, effect-to-cause, cause-to-effect with intervention, and effect-to-cause with intervention. By calculating the proportion of cases where models correctly answer all four question types, CausalBench effectively assesses whether LLMs truly understand the underlying causal relationships, mitigating the impact of models coincidentally providing correct answers without causal comprehension.

The dataset encompasses a diverse set of problems spanning textual, mathematical, and coding domains, addressing the limitations of existing causal reasoning benchmarks. Evaluated on Causal-Bench, state-of-the-art LLMs demonstrate stronger performance on mathematical problems compared to textual and coding tasks. Notably, models with superior causal reasoning abilities tend to exhibit lower hallucination rates, suggesting a potential link between the two capabilities.

Despite its contributions, CausalBench has several limitations, including the need for expanded domain coverage and deeper exploration of the intrinsic mechanisms connecting causal reasoning and hallucination reduction. Future work will focus on addressing these limitations, further refining the evaluation metrics, and providing insights to advance the development of causal reasoning abilities in large language models. CausalBench serves as a robust tool and an important step towards achieving artificial general intelligence.

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

## 8 Appendix A: CausalBench Dataset Link

`https://huggingface.co/datasets/CCLV/CausalBench`

## 9 Appendix B: Text Question Example

**Causal Scenario:**

Imagine a self-contained, hypothetical world with only the following conditions, and without any unmentioned factors or causal relationships: Parents' intelligence has a direct positive effect on parents' social status and child's intelligence. Other unobserved factors has a positive direct effect on parents' social status and child's intelligence.

**Question 1:**

If a child is intelligent, would it be more likely that this child had intelligent parents?

**Question Type:**

Inference from Effect to Cause without Intervention

**Ground Truth:**

Yes

**Explanation:** The probability of the child being intelligent given that the parents are intelligent is higher than the probability of the child being intelligent given that the parents are unintelligent, so the ground truth is yes.

**Question 2:**

If the parents are intelligent, is the child more likely to be intelligent?

**Question Type:**

Inference from Cause to Effect without Intervention

**Ground Truth:**

Yes

**Explanation:** The probability of the child being intelligent given that the parents are intelligent is higher than the probability of the child not being intelligent given that the parents are intelligent, since parent's intelligence has positive effect on child's intelligence.

**Question 3:**

If we intervene to make the parents intelligent (e.g., through education or training), is the child more likely to be intelligent?

**Question Type:**

Inference from Cause to Effect with Intervention

**Ground Truth:**

Yes

**Explanation:** By intervening to increase the parents' intelligence, the child's intelligence is more likely to increase due to the causal chain. Although other unobserved factors also affect the child's intelligence, the direct positive effect of parents' intelligence still exists.

**Question 4:**

If we observe a child is intelligent, and then intervene to make the child unintelligent (e.g., through some kind of impairment), does this make it less likely that the child's parents are intelligent?

**Question Type:**

Inference from Effect to Cause with Intervention

**Ground Truth:**

582 No

**Explanation:**

584 The child's intelligence is the result of the combined effects of parents' intelligence and other factors.
585 Even if we intervene to decrease the child's intelligence, it does not change the parents' level of
586 intelligence. Therefore, in this case, the change in the child's intelligence does not affect our judgment
587 of whether the parents are intelligent or not.

# 10 Appendix C: Code Question Example

**Causal Scenario:**

```
class SalesData {
    int totalSales, newSubscriptions;
    double pricePerSubscription;

    public SalesData(int newSubscribers, double price) {
        this.newSubscriptions = newSubscribers;
        this.pricePerSubscription = price;
        updateSales();
    }

    public void updatePrice(double newPrice) {
        this.pricePerSubscription = newPrice;
        updateSales();
    }

    public void addSubscriptions(int additionalSubs) {
        this.newSubscriptions += additionalSubs;
        updateSales();
    }

    private void updateSales() {
        totalSales = (int)(newSubscriptions * pricePerSubscription);
    }

    public int getTotalSales() {
        return totalSales;
    }
}

SalesData monthlyReport = new SalesData(100, 10.0);
monthlyReport.addSubscriptions(50);
monthlyReport.updatePrice(15.0);
```

**Question 1:**

623 If the number of new subscriptions increases, will total sales also increase, assuming no other
624 changes?

**Question Type:**

626 From cause to effect without intervention

**Ground Truth:**

628 Yes

**Explanation:**

630 The method 'addSubscriptions' adds new subscriptions and then immediately calls 'updateSales',
631 which recalculates total sales based on the new number of subscriptions and the current price per

subscription. Therefore, with no other changes, increasing the number of new subscriptions directly leads to an increase in total sales.

**Question 2:**

Does an increase in total sales imply an increase in the price per subscription?

**Question Type:**

From effect to cause without intervention

**Ground Truth:**

No

**Explanation:**

An increase in total sales can occur either from an increase in the price per subscription or from an increase in the number of new subscriptions due to the calculation in 'updateSales'. Hence, an increase in total sales does not necessarily imply that the price per subscription has increased; it could also be due to an increase in the number of subscriptions.

**Question 3:**

If we manually increase the price per subscription, will this result in an increase in total sales?

**Question Type:**

From cause to effect with intervention

**Ground Truth:**

Yes

**Explanation:**

Increasing the price per subscription using 'updatePrice' method causes 'updateSales' to be called, calculating the new total sales using the increased price. Assuming the number of subscriptions remains constant, this intervention in price directly causes an increase in total sales.

**Question 4:**

If total sales decrease after an intervention, does this mean we decreased the number of new subscriptions?

**Question Type:**

From effect to cause with intervention

**Ground Truth:**

No

**Explanation:**

A decrease in total sales after an intervention could be due to either a decrease in the number of new subscriptions or a decrease in the price per subscription. As these two factors multiply to compute total sales, the decrease could be attributed to either factor independently or both. Thus, a decrease in total sales does not definitively determine that the intervention was a decrease in the number of new subscriptions.

# 11 Appendix D: Math Question Example

**Causal Scenario:**

Investigate the influence of a linear operator transformation $z = L(x)$ on a vector field $x$ governed by $\frac{d}{dt}x = Mx$, where $M$ is a constant matrix. The transformation $L$ represents another linear operator with a constant matrix.

**Question 1:**

If the transformation $z = L(x)$ is applied immediately at $t = 0$ to a vector $x_0$, followed by evolution under $\frac{d}{dt}x = Mx$ without further intervention, does the state $z(t)$ at $t = T$ exactly replicate the result of evolving $x_0$ directly under $\frac{d}{dt}x = Mx$ until $t = T$?

**Question Type:**

From cause to effect without intervention

**Ground Truth:**

No

**Explanation:**

Applying the transformation $z = L(x)$ modifies the initial conditions. The trajectory of $z(t)$ and $x(t)$ would differ unless $L$ commutes with the exponential of $M$, which generally is not the case. Hence, the state transformations under $L$ can produce a distinct evolutionary path in comparison to the direct evolution of $x_0$.

**Question 2:**

Can the original vector $x_0$ be reliably determined at $t = 0$ after observing the vector $z(t)$ at $t = T$, without knowing if the transformation $L$ was applied?

**Question Type:**

From effect to cause without intervention

**Ground Truth:**

No

**Explanation:**

Without information on the application of $L$, reconstructing the exact initial state $x_0$ from $z(t)$ is not straightforward. The application of $L$ can alter the vector in ways that are not easily reversible, especially if $L$ and $M$ are not designed to reveal their effects straightforwardly.

**Question 3:**

If an additional linear transformation $H$ is applied at time $t_1$ as an intervention, will the final state $z(T)$ at $T > t_1$ be independent of the initial transformation $L$ and solely determined by $M$ and $H$?

**Question Type:**

From cause to effect with intervention

**Ground Truth:**

No

**Explanation:**

The final state $z(T)$ will depend on $L$, $M$, and $H$. The transformations imposed by $L$ initially, and $H$ later, both play critical roles. These factors, combined with the dynamics driven by $M$, contribute to a state at $T$ that relies on all three matrices, affected by their interaction and properties.

**Question 4:**

Based on knowing only the vector $z(T)$ at time $T$, is it feasible to precisely identify the transformations (L, H, or both) that were previously applied?

**Question Type:**

From effect to cause with intervention

**Ground Truth:**

No

**Explanation:**

Determining which transformations were applied based on the final vector $z(T)$ alone is challenging due to the overlapping effects matrices may have in transforming the state space. The interactions of $L$

and $H$ with the matrix exponential of $M$ can result in equivalent states from different transformation sequences.

