# OpenReview forum: "CausalBench: A Comprehensive Benchmark for Evaluating Causal Reasoning Capabilities of Large Language Models"
_NeurIPS.cc/2024/Workshop/MATH-AI — MATH-AI 24_

### Official Review · Reviewer_PKJJ · 2024-10-06
**A Comprehensive and Rigorous Benchmark for LLM Causal Reasoning, Needing Condensation for Workshop Fit**

**Rating:** 7
**Confidence:** 3

**Review:**

The paper presents CausalBench, a comprehensive benchmark for evaluating causal reasoning in LLMs across multiple domains (text, math, and code). The benchmark's breadth and depth, as well as the rigorous methodology used to create and validate the dataset, make it a valuable contribution to AI research on causal reasoning.

Strengths
- Comprehensive Benchmark Coverage: CausalBench covers a wide range of causal reasoning tasks, including cause-to-effect, effect-to-cause, and interventions, across diverse domains. This makes it a well-rounded benchmarks available for assessing causal reasoning in LLMs.
- Robust Dataset Creation and Quality Control: The dataset is carefully designed using a three-step process: manual generation, scaling through LLMs, and quality control by human experts and causal inference engines. This ensures the dataset’s accuracy, diversity, and reliability for evaluating causal reasoning.
- Thorough Evaluation Across Models: The paper provides empirical evaluations of several state-of-the-art LLMs (e.g., GPT-4, Claude-3), offering insights into their strengths and limitations in causal reasoning tasks. This comprehensive analysis enhances the paper’s practical relevance and provides a useful benchmark for model performance.

Area for Improvement
- Too long for the workshop: While the paper’s comprehensive nature is a strength, its length and level of detail make it exceeds the 4 page limit of this workshop. Some sections, particularly around dataset creation and evaluation results, could be condensed and move to Appendix.

---

### Official Review · Reviewer_CkZ9 · 2024-10-07
**Minor issues but a step in the right direction**

**Rating:** 6
**Confidence:** 4

**Review:**

**Summary:** The paper introduces a new benchmark dataset for causal reasoning in LLMs: *CausalBench*. The dataset is aimed at addressing the limitations of existing causal reasoning benchmarks by expanding to multiple domains (text -> text / code / math) and measuring causal reasoning via a comprehensive set of 4 dimensions (cause-to-effect and effect-to-cause with and without intervention) rather than simple cause-to-effect. The paper scales up an initial curated dataset using a combination of LLM bootstrapping combined via cross-verification via a causal inference engine (with a final layer of human in the loop).

**Strengths:**
* **Well-Motivated Approach and a Step Forward:** The expansion to four dimensions of causal understanding: cause-to-effect and effect-to-cause, both with and without interventions is well motivated. Rather than relying solely on commonsense-based causality, the approach integrates elements of causal reasoning to measure if LLMs grasp the underlying causal relations. The human-in-the-loop algorithm for scaling up the dataset by bootstrapping via LLMs and cross-verifying against a causal inference engine is also a solid addition. Combined with the expansion to the code / math domains, the paper thus lays a good foundation for the process of creating more comprehensive causal benchmarks for LLMs.

**Weaknesses:**
* **Benchmark fundamentally has many of the same limitations as previous benchmarks:** The benchmark hopes to better distinguish if a model understands the causal model as opposed to just guessing / knowing the answer. The yes / no ground truth carries the same limitation that even if a model gets an answer right, it doesn’t mean it grasps the underlying causal model. Also as a public benchmark, the questions will inevitably make their way to the training data of major LLMs and the benchmark design doesn’t guard against leaking.
* **Benchmark evaluation requires more thorough analysis:** The performance relations between the four dimensions is understudied. E.g in Table 3 (math benchmark evals), many models show significantly higher performance on questions with intervention (a harder task) as opposed to without intervention *which could point to fundamental issues in the intervention dataset design.* This also applies to other benchmarks (text / code) where the performance on with intervention questions is similar to without intervention.
* **Preference Leakage from LLM Bootstrapping:** The bootstrapping procedure depends on expansion via GPT-4 Turbo which adds the risk of inductive bias / preference leakage from that specific LLM (the generated dataset will be more in-distribution to GPT-4 Turbo than other LLMs). While all expansion procedures (including humans) are prone to inductive biases, using the same model that we also want to evaluate contaminates both the experiments and the benchmark to a higher degree. Bootstrapping via LLMs is not a bad idea, but it would be good to either mitigate this bias or at least have some measure of possible contamination in evaluations.

**Questions:**
* **Why not leverage explanation in Ground Truth:** The yes / no answers can be guessed right with a 50% chance and getting a right answer doesn’t mean the model grasps that specific underlying causal model relation. Given the dataset already contains the explanation behind the ground truth, have you considered other approaches (e.g. multi-choice question formats) to leverage this ?
* **Mistral 7B Results on Math:** Is there any explanation around these results ? Not only are the numbers suspiciously high, the model does much better on questions involving intervention rather than those without intervention. This is also true for other models (e.g. Gemma 7B goes from ~50% to 90%). Could it be that somehow intervention leaks information for the math questions ?
* **How do you handle scenarios where both GPT-4 Turbo and Causal Inference Engine get an answer wrong:** The human in the loop only triggers when the two paths disagree. But if they agree wrongly, is there some process to handle it ?
* **How do you handle subjectivity introduced by interventions:** Interventions can force the model to use assumptions outside the problem specifications e.g. in the parent-child example in Appendix B, Question 3 and 4 assume that intelligence relation is determined at birth rather than being dynamic (fair common sense assumption, just not specified in the question). Since in these scenarios, the model now has to rely on its own world knowledge to answer the questions, does this affect the efficacy of your benchmark and if yes, how do you suggest mitigating it ?

**Suggestions:**
* **May want to revisit the name CausalBench:** Multiple papers that introduce benchmark of the same name ([Zhou et al](https://arxiv.org/pdf/2404.06349), [Chevalley et al](https://arxiv.org/abs/2210.17283))
* **Sort / Highlight results in table 1, 2 and 3:** Could you please either sort the results or bold the best ones per column.
* **Add plot of reasoning benchmark vs hallucination rate:** For section 5, could you also please plot a graph to map performance on your benchmark (e.g. text) against the hallucination rate to better visualize correlation between the two.
* **Add analysis on the performance across different causality aspects:** It might be valuable to map the performance relation between models across the 4 aspects to establish benchmar’s validity. In an ideal setup, adding an intervention should reduce the model’s accuracy (which doesn’t happen in the current evals). Also understanding how cause-to-effect and effect-to-cause accuracy varies would shed light on any intrinsic properties of your dataset.

---

### Official Review · Reviewer_Kb4L · 2024-10-07
**A solid contribution for evaluating causal reasoning capabilities of models and the correlation with hallucinations**

**Rating:** 8
**Confidence:** 3

**Review:**

### Clarity

#### Pros
- The paper is structured well, with the authors ensuring that all important sections are included the corresponding context is sufficiently explained in each section.
- The use of an explanatory diagram for steps involved for evaluation dataset construction was helpful, as a reader, to understand the overall process.
- Appreciate a packed appendix providing an example each for the text, math and code domain problems.

#### Cons
- A direct typo on Page 4, paragraph 2, in the first sentence, where _categorized_ was used instead of _categorize_. The exact sentences was "_we can categorized the original questions..._"
- Some sections were quite dense and could use diagrams/tables. For example, section 3.4.1 where the authors explain the design of the causal inference engine, could have a few visual flow diagrams for a more vivid description of the steps between input and output.

### Quality

#### Pros
- There is a rigorous methodology to evaluate the performance of the models in response to the questions on CausalBench.
- The additional effort to correlate the performance of the models on the benchmark with the hallucination leaderboard is useful.
- The authors sufficiently address the limitations and the potential of future expansion of the current work.

#### Cons
The evaluation metrics used on the benchmarks are derivative and may not be comprehensive. There may be a need to design domain-specific metrics for such a benchmark.
- There was no clear discussion about removing bias between using the same model variants (GPT-4) for the dataset construction and for final experiments using the benchmark.
- It would have been helpful for the authors to compare the performance of the models with the existing benchmarks, to emphasize the importance of their contribution.

### Originality

#### Pros
- The benchmark brings novelty through its contribution, especially with the four perspectives of cause-to-effect and effect-to-cause, with and without intervention.
- The dataset and the underlying questions, widely categorized by four perspectives, which is significant.

#### Cons
- This benchmark derives from existing work, in particular for the data matching and casual effect estimation (section 3.4.1) under the casual inference engine design.
- The paper does not present new/seminal contributions to metrics for evaluating multi-domain causal reasoning.

### Significance

#### Pros
- The paper addresses a critical aspect of advancing LLM capabilities.
- A benchmark for evaluating the general reasoning capabilities of models across domains, is indeed a important step towards AGI. More importantly, this is key to being able to employ models for better productivity across general tasks.

#### Cons
- The paper does not provide a clear direction of follow-up research for readers to pick up on, which may limit its significance beyond this work.

---

### Decision · Program_Chairs · 2024-10-07

Accept